# Longitudinal Immunoprofiling of the CD8^+^ T-Cell Response in SARS-CoV-2 mRNA Vaccinees and COVID-19 Patients

**DOI:** 10.3390/vaccines13060551

**Published:** 2025-05-22

**Authors:** Jesús Emanuel Brunetti, Beatriz Escudero-Pérez, Fátima Lasala, Gonzalo Rivas, Mikel Mancheño-Losa, David Rial-Crestelo, Jaime Lora-Tamayo, Dániel Cadar, Miles Carroll, Rafael Delgado, César Muñoz-Fontela, Estefanía Rodríguez

**Affiliations:** 1Bernhard Nocht Institute for Tropical Medicine, 20359 Hamburg, Germany; jesus-emanuel.brunetti@inserm.fr (J.E.B.); beescudero@clinic.cat (B.E.-P.); daniel.cadar@bnitm.de (D.C.); munoz-fontela@bnitm.de (C.M.-F.); 2Centre de Recherche de Saint Antoine (CRSA), Institut National de la Santé et de la Recherche Médicale (INSERM), Sorbonne Université, 75012 Paris, France; 3German Center for Infection Research (DZIF), Partner Site Hamburg-Borstel-Lübeck-Riems, 38124 Braunschweig, Germany; 4Department of Clinical Microbiology, Hospital Clinic de Barcelona, 08036 Barcelona, Spain; 5Barcelona Institute for Global Health (ISGlobal), Hospital Clinic de Barcelona, Universitat de Barcelona, 08036 Barcelona, Spain; 6Instituto de Investigación, Hospital Universitario 12 de Octubre (Imas12), 28041 Madrid, Spain; flasala.imas12@h12o.es (F.L.); gonzalo.rivas@salud.madrid.org (G.R.); rafael.delgado@salud.madrid.org (R.D.); 7Department of Internal Medicine, Hospital Universitario 12 de Octubre, 28041 Madrid, Spain; mikel.mancheno@salud.madrid.org (M.M.-L.); david.rial@salud.madrid.org (D.R.-C.); jalorata@ucm.es (J.L.-T.); 8CIBER de Enfermedades Infecciosas (CIBERINFEC), Instituto de Salud Carlos III, 28029 Madrid, Spain; 9Nuffield Department of Medicine, University of Oxford, Oxford OX3 7BN, UK; miles.carroll@ndm.ox.ac.uk; 10School of Medicine, Universidad Complutense de Madrid, 28040 Madrid, Spain; 11National Centre for Microbiology, Instituto de Salud Carlos III, 28220 Madrid, Spain

**Keywords:** SARS-CoV-2, CD8^+^ T cells, T-cell activation, T-cell memory, TCR, COVID-19, natural infection, vaccination

## Abstract

**Background**: SARS-CoV-2 was the causing agent of the COVID-19 pandemic, which resulted in millions of deaths worldwide and massive economic losses. Although there are already several vaccines licensed, as novel variants develop, understanding the immune response induced by vaccination and natural infection is key for the development of future vaccines. **Methods**: In this study, we have used flow cytometry and next-generation sequencing to assess the longitudinal CD8^+^ T-cell response against natural infection and vaccination in convalescent and vaccinated individuals, from early activation to immune memory establishment. Moreover, we have characterized the T-cell receptor clonality and diversity at different stages post-infection and post-vaccination. **Results**: We have found no significant differences in CD8^+^ T-cell activation during the first three weeks post-infection compared to the first three weeks after first vaccination. Conversely, natural infection resulted in sustained high levels of T-cell activation at four weeks post-infection, a point in which we observed a decline in T-cell activation post-vaccination despite boosting with a second vaccination shot. Moreover, additional vaccination did not result in enhanced T-cell activation. Of note, we have observed variations in the memory subset structure at every stage of disease and vaccination. Overall, both infection and immunization induced a highly diverse T-cell receptor repertoire, which was observed both between study groups and between patients inside a given group. **Conclusions**: These data contribute to expand our knowledge about the immune response to SARS-CoV-2 infection and vaccination and call for additional strategies to enhance T-cell responses by booster immunization.

## 1. Introduction

The *Coronaviridae* family is composed of positive-sense single-stranded RNA-enveloped viruses that cause respiratory tract infections in humans and other mammals [1]. The highly pathogenic human viruses known so far in this family belong to the *Beta-Coronavirus* genus and include the severe acute respiratory syndrome coronavirus (SARS-CoV), the Middle East respiratory syndrome coronavirus (MERS-CoV), and severe acute respiratory syndrome coronavirus type 2 (SARS-CoV-2) [1]. SARS-CoV-2 is the etiological agent of coronavirus disease 19 (COVID-19) that caused the latest coronavirus pandemic. It appeared for the first time in Wuhan, China [2], and spread globally, challenging the healthcare systems worldwide. The clinical symptoms of infection caused by SARS-CoV-2 range greatly among patients, resulting in asymptomatic infections as well as severe and fatal cases. As of 10 November 2024, the World Health Organization (WHO) has reported 776,841,264 COVID-19 cases and 7,075,468 deaths worldwide [3]. Vaccination efforts have helped to diminish the number of severe cases [4,5,6]; however, it has been demonstrated that neutralizing antibody activity against SARS-CoV-2 generated after immunization diminishes over time, requiring repeated rounds of vaccination [7,8,9,10,11]. Although the WHO has declared the end of the pandemic in May 2023 [12], infection peaks occur frequently [13] as new variants appear. Despite vaccination efforts, these infections still cause high hospitalization rates and deaths among the elderly and patients with comorbidities, posing a serious public health threat [14,15,16,17].

Upon viral infection, the activation of virus-specific CD8^+^ T lymphocytes plays an important role in viral clearance and the generation of antigen-specific memory T cells [14]. Antigen-presenting cells (APCs) such as dendritic cells and macrophages engulf and process viral antigens coming from infected cells. These antigens are processed into specific peptide fragments that bind to MHC molecules displayed on the surface of these APCs. T-cell receptors (TCRs) in T lymphocytes recognize the MHC–peptide complexes and interact with them in a very specific manner through their variable region. This interaction induces the activation of T cells through different cellular pathways, triggering the cellular immune response [15]. The diversity of the TCR, which determines its ability to recognize a wide range of antigens, arises during T-cell development by somatic recombination of distinct V and J genes for the alpha chain and V, D, and J genes for the beta chain. This recombination forms hypervariable regions or complementary-determining regions (CDRs) in the TCR. One of the highest hypervariable regions that determines the diversity of the TCR in the context of viral antigen recognition is CDR3, which is commonly used to evaluate TCR diversity [16,17]. When naive T cells meet a cognate antigen–MHC complex through their TCR, a signaling cascade is triggered, inducing their proliferation and development into activated effector T cells, which can eliminate infected cells. The diversity and clonality of the TCR repertoire determine the efficacy to respond not only to a primary infection but also to a second one [18]. The evaluation of the diversity and clonality of T-cell populations through TCR sequencing allows us to track T-cell clones over time and analyze them through the expansion and contraction phases of the immune response. Previous studies have shown that the TCR repertoire can be modulated upon SARS-CoV-2 infection. For example, it has been shown that during early infection in severe cases, there is a reduction in the diversity of the TCR, which returns to baseline levels upon recovery. This is related to changes in the expression of T-cell genes involved in immune homeostasis. In addition, other studies have observed the same TCR clonotype pattern among patients or similar usage of the V and J genes in a specific disease stage, suggesting common SARS-CoV-2 immune signatures depending on disease severity and phase [19,20,21,22,23,24]. However, it is not known if TCR repertoires are comparable in natural immunization and vaccination and whether differences in TCR repertoires are informative about the quality of the T-cell response, in particular the transition between primary and memory CD8^+^ T-cell responses, which are protective against reinfection [25,26,27,28].

Here, we have analyzed and compared the CD8^+^ T-cell response to SARS-CoV-2 infection in severe COVID-19 patients and in vaccinees that received mRNA-based vaccines (BNT162b2 from Pfizer/BioNTech and mRNA-1273 from Moderna). We describe the activation kinetics and the memory profiles in both types of immunization and explore the clonality, diversity, and V/J gene usage of the CDR3 region in SARS-CoV-2-specific CD8^+^ T cells. These analyses provide insights into the dynamics and phenotype of memory T cells in the early and late phases of the immune response.

## 2. Materials and Methods

### 2.1. Cohorts’ Description

Samples from hospitalized COVID-19 and convalescent patients were obtained between December 2020 and March 2022 through a collaboration with the “Hospital Universitario 12 de Octubre” in Madrid (Spain) after obtaining informed consent (Project ECRIN, IRB Reference ECRIN-C19-HU12O). Symptomatic patients were admitted to the hospital after having a positive COVID-19 test and severe respiratory symptoms such as respiratory distress and low oxygen saturation. SARS-CoV-2 detection was made using RT-PCR on the E gene in nasopharyngeal swabs and the Panther Fusion Hologic (San Diego, CA, USA) automated molecular diagnostic platform, as previously described [29]. Patients reported suffering symptoms between 2 and 10 days before the hospital visit. Based on this and the incubation period for SARS-CoV-2, the mean time for virus exposure was set at 7 days before hospital admission, which was defined as day 0 post-infection. Blood samples were obtained at the time of diagnosis and twice weekly during hospitalization. Upon discharge from the hospital, samples were obtained every 3 months for up to 18 months. For these studies, we used samples from 21 different patients, from which 2 to 5 samples were harvested over time to assess the kinetics of the immune response. A total of 81 samples were collected and used for the studies presented here. Patients were classified into different severity categories based on reference [30]. Category 4 corresponded to hospitalized patients that did not need extra oxygen supply but still required continuing medical care specific for COVID-19; category 5 corresponded to hospitalized patients that needed extra oxygen supply; and category 6 corresponded to hospitalized patients on high-flow oxygen devices or non-invasive ventilation systems. The medical records of these patients include data on demographics, oxygen levels, COVID-19 severity, and therapy applied.

Samples from mRNA vaccinees were obtained from a cohort of healthy volunteers at the Bernhard Nocht Institute for Tropical Medicine (BNITM) in Hamburg (Germany) with approved consent. The protocol for the use of these samples was authorized by the ethics committee of the German Medical Association and the ethics committee of the Ärztekammer Hamburg (PV4780). Samples were harvested between May 2021 and January 2022. For this cohort, samples were harvested one, two, three, and four weeks after the first vaccine shot and then four weeks after the second and third shot, respectively. In total, 41 samples were collected from 10 different volunteers and used for the studies.

Samples collected from both cohorts were classified as belonging to the early or to the late immune response. Samples harvested during the first 3 weeks of being naturally infected or receiving the first vaccination shot were classified within the early immune response group. Samples that were harvested in the fourth week or later upon natural infection or first vaccination or after 4 weeks after receiving the second or third vaccine shot were classified in the late immune response group.

### 2.2. PBMC Preparation

PBMCs were isolated from heparinized peripheral blood by either using a Ficoll–Hypaque gradient (PANcoll, density 1.077 g/mL; PAN Biontech GmbH, Aidenbach, Germany) or red blood cell (RBC) Lysis Buffer 1X (BioLegend, San Diego, CA, USA), depending on the origin.

The isolation with a Ficoll gradient was performed as described before [31]. The isolation by using the Lysis Buffer (Biolegend (San Diego, CA, USA) was performed following the manufacture guidelines.

Following isolation, PBMCs were frozen in freezing media (fetal bovine serum and 10% DMSO) and stored in liquid nitrogen until use.

Before antibody staining, PBMCs were thawed, washed with PBS 1X, and then resuspended in 200 μL of a staining solution (SB) containing 1X PBS, 2 mM MgCl_2_, and DENARASE^®^ (c-LEcta GmbH, Leipzig, Germany).

### 2.3. Flow Cytometry

For flow cytometry analysis, PBMCs were first stained for 20 min at room temperature (RT) with the same MHC class I tetramer labeled with two different fluorophores, APC or PE. The tetramer was based on the peptide YLQPRTFLL from the SARS-CoV-2 spike protein (Cat WB05824, Immudex, Copenhagen, Denmark). After centrifugation for 5 min at 2000 rpm and 4 °C, the supernatant was removed, and PBMCs were incubated with Zombie UV (dilution 1:1000) for 10 min at RT and then blocked with FC True Staining Blocking (dilution 1:100) for 20 min. After this time, cells were centrifuged at 2000 rpm for 5 min at 4 °C, and the supernatant was removed, and cells were stained with the following antibody cocktail for 1 h at RT: CD19-FITC, CD56-FITC, CD3-BV421, CD4-BV711, CCR7-APC.CY7, CD45RA-BV605, CD8-BV510, HLA-DR-PE.Cy7, and CD38-PerCP.Cy5. Next, PBMCs were fixed with BD Cytofix (BD Bioscience, Heidelberg, Germany) for 15 min at RT in the dark and then centrifuged and resuspended in 200 μL of PBS 1X for further analysis. All antibodies, life/death dyes, and blocking solutions were bought from Biolegend (San Diego, CA, USA). A schematic of the gating strategy is shown in Appendix A.

Samples were analyzed in a BD LSR Fortessa cytometer (BD Bioscience, Heidelberg, Germany). Data were processed using the FlowJo10.5 software.

### 2.4. Sample Preparation for Bulk T-Cell Receptor (TCR) Sequencing

Thawed PBMCs were stained for cell sorting using SARS-CoV-2 tetramers, as described above, and the following antibodies were used: CD19-FITC, CD56-FITC, CD3-BV421, CD4-BV711, and CD8-BV510. Cells were sorted using a BD FACSAria III cell sorter (BD Bioscience) to obtain SARS-CoV-2-specific cells (CD8^+^ Tet^+^ cells).

Total RNA was extracted from sorted cells by using the ReliaPrep^TM^ RNA Cell Miniprep System (Promega GmbH, Walldorf, Germany), following the manufacturer’s guidelines. Reverse transcription was performed using random primers and the iScript cDNA Synthesis Kit (Bio-Rad, cat 708891, Feldkirchen, Germany).

### 2.5. TCR Amplification

TCR amplification was performed as previously described [32]. Briefly, the TCR β complementarity-determining region 3 (CDR3) locus was amplified by using 1 μL of cDNA and the QIAGEN multiplex PCR kit, following the manufacturer’s guidelines. A total of 45 forward primers targeting functional TCR Vβ sequences and 13 reverse primers targeting functional TCR Jβ segments were pooled and used for the first PCR amplification round [17]. This set of primers had overhang adapters that served as the specific target for the primers of the second PCR amplification round. In this second reaction, the Illumina Nextera XT dual-index primers, which served as sequencing adapters, were added. After each PCR round, the products were purified using Agencourt AMPure XP beads (Beckman Coulter GmbH, Aachen, Germany), following the manufacturer’s instructions, to recover amplicons of around 200 base pairs (bps). The size of the fragments was checked through agarose gel electrophoresis. The primer sequences and the full detailed protocol has been previously described [17].

### 2.6. Library Preparation and Sequencing

The concentration of purified amplicons was measured using the Roche qPCR-KAPA library quantification kit (Roche, Mannheim, Germany), and the size distribution was assessed in the Agilent Bioanalyzer 2100 using the DNA 1000 kit (Agilent, Waldbronn, Germany). The final library was built by pooling samples in equal parts. The pooled and denatured TCR library was diluted to a final concentration of 12 pM and spiked with 5% PhiX and then sequenced on a NextSeq 550 platform using a NextSeq 500/550 Mid Output Kit v2.5 (Illumina, San Diego, CA, USA) with 150 cycles (2 × 75 bp paired-end sequencing).

### 2.7. Bioinformatical Analysis of Sequencing Results

FASTQC was first used to test sequence read quality on the Galaxy online public platform Galaxy.eu server (usegalaxy.eu accessed on 16 March 2023) [33]. The matching sequence readings were trimmed and filtered for quality on the same web server. To construct complete sequences, read pairs with a minimum quality score of 20 were merged throughout their overlapping sequences using Galaxy. TRUST4 [34] was used to align the obtained sequences with the reference V, D, and J TCR genes to identify which genes were present in each sample. Then, identical CDR3 sequences were identified and classed as clonotypes. All this information was finally exported as a tab-delimited, readable text file. CDR3 sequences with one count were discarded from the following analysis to avoid possible errors during the sequencing steps. The files collected via TRUST4 were then subjected to sophisticated data analysis using the R package Immunarch (version 0.9.1) [35].

### 2.8. Statistical Analysis

Statistical analyses were performed using the R (version 4.4.0, 2024-04-24 ucrt, for Windows X64) [36] and RStudio (version 2024.09.1 +394 for Windows X64) [37] software. To compare the activation and memory profiles over time and within groups, the data were analyzed through generalized least square (GLS) models with dummy variables, when necessary, followed by Tukey’s multiple comparison test. Models’ assumptions were checked.

The non-parametric Kruskal–Wallis test was used to compare the usage of the V and J gene families between groups, followed by pairwise Wilcoxon tests with multiple testing corrections. (The *p*-adjust method was Benjamini–Hochberg.)

The statistical significance levels are as follows: * *p* < 0.05, ** *p* < 0.01, *** *p* < 0.001, and **** *p* < 0.0001.

Data are presented as the mean ± standard deviation (SD).

TCR diversity was measured using the Gini–Simpson index in Immunarch (version 0.9.1). The Jaccard index was used to determine the overlap of TCR datasets. Immunarch was used to describe the TCR clonal space by calculating the percentage of the top ten clonal proportions, as well as to describe gene usage in each group and track clonotypes over time for different patients.

## 3. Results

### 3.1. Early Immune Response to SARS-CoV-2 Infection or Vaccination Showed Low and Similar CD8^+^ T-Cell Activation

To analyze the early CD8^+^ T-cell response to natural SARS-CoV-2 infection (Nat) or vaccination (Vac), we obtained peripheral blood mononuclear cells (PBMCs) from two different cohorts. The first cohort was composed of healthcare workers from the “Hospital 12 de Octubre” in Madrid (Spain) who were infected with SARS-CoV-2 during the years 2020 to 2022. The second cohort was composed of healthy individuals that have received different doses of the SARS-CoV-2 mRNA vaccine from Pfizer/BioNTech or Moderna. Some of these donors were diagnosed with SARS-CoV-2 infection at later time points in the study.

Following previous studies [21,22,23,38,39,40], samples were classified as those belonging to either the early immune response or to the late immune response. Within the early immune response sample group, we included samples obtained between 1 and 3 weeks, either after natural infection (Nat-early, total number of samples in this group (n_total_) = 21, and age average = 57.67 ± 15.29 years old) or 1–3 weeks after receiving the first vaccination dose (Vac-1x, n_total_ = 7, and age average = 37.14 ± 13.41 years old). In these samples, we studied the activation profiles of CD8^+^ T cells by measuring the percentage of cells co-expressing the CD38 and HLA-DR markers using flow cytometry.

Our results showed that during the first week of the early immune response, SARS-CoV-2 natural infection (Nat-early) induced, on average, almost a 3-fold higher percentage of activated CD8^+^ T cells (27.26 ± 21.94%, average ± stdv) compared to vaccinees that have received the first vaccine dose (Vac-1X) (9.54 ± 8.23%); however, when we applied a GLS model followed by Tukey’s multiple comparison test, we observed that this difference was not statistically significant, probably due to variations inside the groups and the low number of samples analyzed. During the second and third weeks of the early immune response, the difference in the percentage of activated CD8^+^ T cells between Nat-early and Vac-1x decreased (week 2: Nat-early = 21.90 ± 21.08% and Vac-1X = 13.64 ± 11.46%; week 3: Nat-early = 18.30 ± 10.71% and Vac-1X = 12.13 ± 11.39%) (Figure 1).

### 3.2. Late CD8^+^ T-Cell Activation and Memory Profiles Showed Significant Differences Among SARS-CoV-2-Infected Patients and Vaccinated Individuals

Samples from patients of the same cohorts as in Section 3.1 were harvested at further time points. Samples from naturally infected patients that were collected on week 4 post-infection or later were considered in the convalescence phase of the infection and belonged to the late immune response group. Convalescent patients were divided into two sub-groups: early convalescence (EC, n = 7, 56.14 ± 14.80 years of age), where samples were collected between 4- and 10-weeks post-infection, and late convalescence (LC, n = 34, 54.87 ± 15.74 years of age), where samples were collected later than 10 weeks. Those convalescence patients who have additionally received one or two vaccination doses after suffering natural infection were included in a third category of convalescent and vaccinated patients (C/V, n = 15, 65.58 ± 13.92 years of age). Among the vaccinated cohort that belonged to the late immune response group, we included samples obtained 4 weeks after receiving the first vaccination dose (Vac-1X, n = 15, 37.63 ± 12.99 years of age), the second vaccination dose (Vac-2X, n = 11, 38.27 ± 12.81 years of age), or the third one (Vac-3X, n = 4, 43.75 ± 14.93 years of age). Double- and triple-vaccinated individuals received the second and third shots more than three weeks after the first or second dose, respectively.

To investigate the activation profiles at these late stages post-infection or post-vaccination, we compared the percentage of CD8^+^CD38^+^ and HLA-DR^+^ T cells among the groups (Figure 2A). EC patients showed the highest percentage of activated CD8^+^ T cells (24.14 ± 26.48%) compared to LC (5.05 ± 7.18%, with * *p* < 0.05), C/V (2.95 ± 2.96%, with * *p* < 0.05), Vac-2X (2.11 ± 1.47%, with ** *p* < 0.001), and Vac-3X (1.27 ± 0.49%, with * *p* < 0.05) (Figure 2A), which was statistically significant after applying the GLS model followed by Tukey’s test. Among vaccinees, individuals that received one dose (Vac-1X, 13.71 ± 13.78% of activated CD8^+^ T cells) showed significant differences (** *p* < 0.001) compared to Vac-2x (2.11%) and Vac-3X (1.27 ± 0.49%).

By comparing the activation profiles between the early and late stages of the immune response, we could observe that SARS-CoV-2 infection mildly increased the activation of CD8^+^ T cells from the third week of natural infection (Nat-early) to the early convalescence phase (EC) (from 18.3% to 24.14%) and then greatly decreased again in the late convalescence phase (LC) (5.05%). Conversely, vaccination caused discrete CD8^+^ T-cell activation that was lower compared to the one observed upon natural infection, and this activation did not further increase over time or after the second and third vaccination shots.

During viral infection, CD45RA and CCR7 are markers that reflect T-cell antigen experience and homing to lymphoid tissues, respectively, which are important for characterizing T-cell memory subtypes. These indicators help in determining the quality of the T-cell response at different stages of the disease. In addition, these markers are used to differentiate functionally distinct CD8^+^ T cells based on their homing potential, activation history, and effector capabilities [41]. To analyze the proportion of circulating memory CD8^+^ T-cell subsets in naturally infected individuals and compare it to vaccinees, we measured using flow cytometry the percentage of the CD45RA and CCR7 markers in the surface of CD8^+^ T cells and classified them into four categories: naive cells (CD45RA^+^CCR7^+^), which are unexposed cells to antigens that home to the lymph nodes; central memory or CM cells (CD45RA^−^CCR7^+^), which have been exposed to antigens and retain memory and can proliferate and home to the lymph nodes; effector memory or EM cells (CD45RA^−^CCR7^−^), which are effector cells with immune memory and involved in the peripheral immune response; and terminally differentiated CD45RA-positive effector memory or EMRA cells (CD45RA^+^CCR7^−^), which are terminally differentiated and cytotoxic memory T cells that have been related to senescent or exhausted phenotypes [42].

The analysis of the memory T-cell profile in our cohorts showed statistically significant differences between groups. For example, among the convalescent patients, the percentage of EMRA cells was higher in LC and C/V patients than in EC (Figure 2B), while naive (Figure 2C) and CM cells (Figure 2E) were higher in EC patients compared to the other two groups. In vaccinees, the proportion of EMRA cells increased after the second vaccination shot and decrease after the third shot (Figure 2B). The percentage of naive cells was low in Vac-1X and Vac-2X and only increased after the third vaccine shot (Figure 2C). CM cells were low in all vaccination groups (Figure 2E), and the percentage of EM cells was very similar among all groups, including convalescence patients (Figure 2D). When we compared the profiles of convalescence and vaccinees, we observed that EMRA cells were higher in vaccinees than in convalescence (Figure 2B), while the opposite was observed regarding CM T cells (Figure 2E). Moreover, the group that included convalescence and vaccinated individuals (C/V) did not show significant differences in the activation and memory profiles when compared to long convalescence (LC), non-vaccinated individuals (Figure 2).

### 3.3. Similar Overall TCR Clonal Diversity Was Observed Among SARS-CoV-2 Convalescence Patients and Vaccinees

The kinetics of the T-cell diversity and clonal expansion throughout infection and vaccination can vary depending on the antigenic source, antigen load, epitope immunodominance, and age of the individual, among other factors [39]. To investigate the T-cell diversity and clonality in representative samples from each infected or vaccinated group, we performed sequencing of complementarity-determining region 3 (CDR3) of the T-cell receptor (TCR) in SARS-CoV-2-specific CD8^+^ T cells. For this analysis, 18 samples that covered all experimental groups from our cohorts were chosen: natural infection-early response (Nat-early, n = 4), early convalescence (EC, n = 4), late convalescence (LC, n = 4), convalescent/vaccinated (C/V, n = 2), and vaccinees (n = 4). In the vaccinees group we included samples from patients with two or three vaccine doses.

PBMCs from these donors were sorted to select CD8^+^ T cells specific for an HLA-A2-restricted epitope derived from the viral spike sequence (YLQPRTFLL), which was recognizable by public CDR3 motifs and identified as the most immunogenic [43]. Sorting was performed by utilizing fluorescently labeled HLA-A2 tetramers bound to the aforementioned peptide (Tet^+^ cells). FACS-sorted cells were then prepared for RNA extraction, cDNA library construction and amplification, and sequencing using the IIlumina NextSeq550 platform.

To analyze TCR diversity, raw sequenced data were first trimmed and filtered for quality with a minimum score of 20 and then analyzed and compared using the Gini–Simpson index (G-S). This method assigns scores from 0 to 1 to evaluate diversity, with 1 assigned to the highest diversity within a set of samples and 0 to samples with the lowest diversity. Figure 3A shows the diversity of the different experimental groups. Vaccinated individuals (Vac) and naturally infected (Nat-early) patients showed the highest Gini–Simpson score, 0.83 ± 0.09 and 0.73 ± 0.38, respectively, and therefore the highest diversity. Diversity decreased in EC (0.51 ± 0.27) and LC (0.60 ± 0.38) samples compared to Nat-early samples. The analysis of samples from two convalescent patients that have received vaccination after recovery showed very different TCR diversity scores. One of the patients, who had received one vaccine dose (C/V-1X), had a Gini–Simpson score of 0.89, while the other, who had received two vaccine shots (C/V-2X), had a score of 0.19. Furthermore, among the vaccinated individuals, which included people receiving two or three vaccination shots, diversity was also high (Gini–Simpson score: Vac-2X = 0.87 ± 0.06 and Vac-3X = 0.71). These results showed that, in general, vaccination induced a similar degree of T-cell diversity as the natural infection. The low diversity observed in the convalescent patient that received two vaccination shots (C/V-2x) could be due to the specificities of that patient, which are discussed below.

Another way to assess the diversity of the TCR repertoire is to measure the length distribution of CDR3. This region, located in the variable region of the TCRβ chain, contains the antigen binding site, and the diversity in length and frequency determines the binding affinity and specificity to the antigen; therefore, it is a way to measure the diversity of the TCR [44].

When we compared the length and frequency of the CDR3 regions between groups (Figure 3B,C), we observed that EC patients had a higher frequency of longer CDR3 regions (a peak of frequency of 39.88% at 60 nt) compared to Nat-early (26.32 ± 46.90% frequency peak at 51 nt) and LC patients (32.91 ± 44.28% frequency peak at 48 nt). Interestingly, the Vac group had the higher frequency peak (27.83 ± 12.51%) at a shorter length (39 nt) compared to the naturally infected groups. Although the length range in the LC and Vac groups was similar to the one in the other groups (lengths from 18 to 72 nt and 18 to 84 nt, respectively), the LC group presented some larger CDR3 regions (~63 nt) that were absent in the Vac group. Interestingly, when comparing LC to C/V-1X or C/V-2X, we observed that while LC and C/V-1X had similar frequencies and distribution of the CDR3 length, the C/V-2X sample had a unique CDR3 length of 33 nt in 98.84% of the cells. This unique length of CDR3 in the C/V-2x group is in concordance with the Gini–Simpson index observed in this patient (Figure 3A), which suggested low diversity.

Through V(D)J recombination, the V, D and J genes rearrange to form different TCRs, and therefore a variety of T-cell clones or clonotypes [17]. This can be measured and quantified by determining the usage of those genes to form the beta (TRB) and alpha (TRA) chains of the TCR.

In this work, we have sequenced the CDR3 region of the TCR beta chain, which allowed us to quantify the usage of the TRBV and TRBJ genes in the different experimental groups (Appendix A, respectively). On the one hand, the TRBV genes with the highest frequency were different in the different experimental groups. For Nat-early patients, TRBV4-1 was the most frequent gene on average (25.50%, range 7.20% to 58.56%), although great differences were found among patients within this group. In the EC group, the TRBV-2 gene was the most frequently used on average (46.21%, range 0.24% to 92.18%), but there was also high intra-group variability. The highest frequently used genes in the LC, C/V-1X, and C/V-2X groups were TRBV6-1 (16.39%, range 6.60% to 35.79%), TRBV6-2 (13.21%), and TRBV7-9 (44.23%), respectively. In the Vac group, genes TRBV6-6 (14.74%), TRBV6-2 (13.86%), TRBV6-1 (12.87%), TRBV 6-8 (11.71%), and TRBV 19 (7.90%) were all highly frequently used (Appendix A). On the other hand, in the analysis of the usage of the TRBJ genes, we saw that the gene TRBJ1-1 was highly frequent in all experimental groups. In addition, TRBJ2 family genes (TRBJ2-1, TRBJ2-3, or TRBJ2-5) were also highly frequent in all groups (Appendix A). These results indicate that the usage of TRBV genes is very different among groups, while the usage of TRBJ genes is more conserved.

To evaluate the TCR similarity between the different experimental groups, we calculated the Jaccard index. This parameter gives a measurement of the overlap between patients’ repertoires, and it has values between 0 (no overlapping) and 1 (full overlapping). Appendix A shows that the overlap between samples was rather low, as only a maximum value of 0.0370 was determined between two Nat-early samples. These results suggest that the TCR repertoire is very different between groups and between patients within a group.

After analyzing the top clonal proportion, we observed that the top 10 more abundant clones occupied 68–82% of the repertoire in the EC, LC, and C/V-2X groups, while they only occupied 48 to 55% of the repertoire in the Nat-early, C/V-1X, and Vac groups (Appendix A).

### 3.4. Kinetics of T-Cell Activation and Clonal Expansion in Convalescent Patients

To better understand individual differences in the kinetics of CD8^+^ T-cell responses after natural immunization vs. vaccination, we sought to determine the kinetics of T-cell activation and clonal expansion over time in three different patients belonging to our cohort of convalescence patients (ECRIN cohort, Spain) from which we harvested samples during the acute infection and the convalescence period. In addition, two of these patients received one or two vaccination doses, respectively, during the late convalescence period.

Patient 1 was a 52-year-old male that during the course of disease needed to be hospitalized and required extra oxygen supply. His disease severity was classified as level 5 based on reference [30]. From this patient, samples were obtained at 14, 42, 70, and 364 days post-infection (PI).

Patient 2 was a 59-year-old male that developed disease severity level 6 [30], requiring non-invasive respiratory assistance (high-flow oxygen devices) and hospitalization. Samples from this patient were harvested at days 15, 74, and 368 PI. At day 97 PI, the patient received a single dose of an mRNA SARS-CoV-2 vaccine (1x).

Patient 3 was a 51-year-old male that presented disease severity level 4 [30] during his stay at the hospital; he did not require oxygen supply but still needed intensive medical care. Samples at days 39, 70, and 364 PI were harvested from this patient. Furthermore, this patient received two vaccination shots at days 130 and 151 PI (2x), respectively.

#### 3.4.1. Activation Profile and Antigen-Specific CD8^+^ T Cells

When comparing CD8^+^ T-cell activation, we observed that patient 1 had very high T-cell activation at 11 days PI (71.84% HLA-DR^+^ CD38^+^ CD8^+^ T cells), which rapidly decreased to 17.62% at day 14 PI (Figure 4A, blue line). This decrease progressed until the end of sampling on day 364, in which no activated CD8^+^ T cells were detected. Moreover, SARS-CoV-2-specific CD8^+^ T-cell (Tet^+^) counts also decreased drastically from day 11 to day 14 PI (0.12% to 0.01%); however, the values stayed stable between days 42 and 70 PI (0.01 and 0.02%, respectively) and then increased on day 196 PI (0.12%) and day 364 PI (0.10%). In line with this, the analysis of the memory T-cell profile showed that at day 14, there were just a few Tet^+^ cells in all memory groups (EM, CM, EMRA, and naive), which increased over time until day 364, where Tet^+^ cells were mainly EMRA and CM T cells (Appendix A, upper panels).

T-cell activation in patient 2 was not as high as in patient 1; however, we saw a similar pattern in which the activated CD8^+^ T cells decreased over time after an initial increase (Figure 4A. patient 2). Antigen-specific T cells were low at the beginning of the disease (less than 0.01% at 8 days PI) and belonged to the EM and CM groups (Appendix A, middle panel). Tet^+^ cells stayed low and became EMRA cells at day 74 until the patient received a vaccination shot at day 97 PI, which greatly increased the percentage of SARS-CoV-2-specific T cells (2.30%) detected at day 368 and allocated them among the EM and CM cells again (Figure 4A, red line, and Appendix A, patient 2).

In patient 3, the activation of CD8^+^ T cells was initially low compared to patient 1, but it was similar to patient 2 (Figure 4A. patient 3, blue line). Nevertheless, it followed a pattern of decrease over time and then rebounded during the convalescence phase. Antigen-specific T cells were rather low and present in all memory groups until day 70 PI, where they were highly picked (from 0.01% to 0.64%) and became EMRA cells, and then decreased after day 161 PI (Figure 4A, patient 3, red line), following a similar pattern to the CD8^+^ T-cell activation curve. At this time point, Tet^+^ cells were mainly EM and CM cells, similar to patient 2 (Appendix A, lower panel). In this case, the two vaccination doses received on days 130 and 151 did not influence T-cell activation or specificity.

#### 3.4.2. TCR Diversity Between Patients

When we analyzed the diversity of the T-cell receptor in the three different patients, we observed that the length of the CDR3 region in SARS-CoV-2-specific T cells changed over time in all patients with some pattern differences (Figure 4B). For instance, patient 1 had mainly clones with CDR3 regions of ~50 nt at 14 days PI (Figure 4B, patient 1), in which the V chains TRBV4-1*01 and TRBV4-3*01, as well as seven other J chains, were mostly used (Figure 5A and Figure 6A, respectively). Over time, the diversity in the length of the CDR3 region increased, and the usage of the V and J genes changed, suggesting that the immune response to SARS-CoV-2 became specific. Every analyzed time point is distinguished by the existence of clones that grew in comparison to the preceding time but shrank in the subsequent time, making the preponderant clones in each time point distinct (Appendix A, patient 1). Lastly, the concept of specialization throughout the disease is supported by the Gini–Simpson score (GS_14d_ = 0.158; GS_42d_ = 0.656; GS_70d_ = 0.604; GS_364d_ = 0.907), which increased over time in patient 1, indicating an increase in diversity.

In patient 2, the CDR3 region presented high diversity in length at all time points PI (Figure 4B, patient 2), and it aligned with a similar Gini score in all of them (GS_15d_ = 0.943; GS_74d_ = 0.837; GS_368d_ = 0.890). In line with this, the usage of the V and J genes was very similar during the course of infection, in which mainly genes from the TRBV28, TRBV27, and TRBV6 families, as well as the TRBJ1-01 gene, were detected (Figure 5B and Figure 6B). Furthermore, a dominating CDR3 clonotype size was identified at every time point, with a notable expansion and subsequent contraction observed at 74 days PI (Appendix A, patient 2).

In contrast to patients 1 and 2, patient 3 showed a unique CDR3 length, which was different at each time point. In addition, this patient had used specific V and J genes at the first time point. In the second and third time points, the usage of the V and J genes increased, but it was still low in diversity compared to patients 1 and 2 (Figure 5C and Figure 6C). Furthermore, patient 3 had a low Gini score at all time points (GS_39d_ = 0.098; GS_70d_ = 0.058; GS_364d_ = 0.191), which indicated lower diversity compared to patients 1 and 2. Moreover, patient 3 presented unique clones at each time point that were not present in the subsequent time point, suggesting a constant growth and shrinkage of distinct clonotypes throughout time (Appendix A, patient 3).

## 4. Conclusions

With millions of deaths and enormous financial losses, SARS-CoV-2’s global expansion has been a severe threat to public health and economies worldwide. Despite the continuous appearance of new variants, vaccination efforts have helped to avoid severe diseases and have alleviated the healthcare systems by decreasing hospitalizations. However, a full understanding of the immune response to infection and vaccination against SARS-CoV-2 and the durability of the immune memory in both cases is still lacking. This is essential for developing new medications, therapies, and diagnostic tools and understanding the level and length of protection upon SAR-CoV-2 infection and vaccination.

In this study, we have performed a longitudinal analysis of T-cell activation and memory profiles by measuring the expression of different markers, as well as performed TCR sequencing of SARS-CoV-2-specific CD8^+^ T cells to profile the cellular immune response in COVID-19 convalescent patients and vaccinated individuals. To analyze the activation of T cells, we have measured the percentage of CD38^+^ and HLA-DR^+^ CD8^+^ T cells, which are markers associated with cell activation [45]. During acute infection, the co-expression of HLA-DR and CD38 is linked to enhanced viral clearance, cytotoxicity, and proliferation [46]. However, the expression of these markers during chronic infection has been associated with immune fatigue, activation-induced cell death, and a loss of function. Moreover, despite their lower activation state, CD38^-^ HLA-DR^+^ CD8^+^ T cells have been associated with a higher capacity to inhibit viral replication and better overall control in a variety of viral infections [45]. In our analyses, the activation of CD8^+^ T cells during the early immune response showed the highest values during the first week of natural infection (Nat-early) compared to the first week in vaccinees (Vac-1x), in which it was rather low. The activation in naturally infected individuals was maintained during the second and third weeks of infection and the early convalescence phase (EC group), with a tendency to decrease that was not statistically significant (Figure 1 and Figure 2A). The activation of CD8^+^ T cells during the early immune response (weeks 1, 2, and 3 post-first vaccine shot) was lower than in the natural infection; however, this difference was not statistically significant. This could be due to the low number of samples analyzed and the high variability between samples within a given group. Interestingly, during the late convalescent phase of COVID-19 disease (LC), the activation of CD8^+^ T cells greatly decreased, and this was maintained after some of these patients received a vaccination dose (C/V). Furthermore, within the vaccinee group, the percentage of activated CD8^+^ T cells was very similar during the first, second, third, and fourth weeks after the first vaccine shot and then declined after the second and third vaccinations (Figure 1 and Figure 2A).

Previous studies have shown that elevated expression of CD38 and HLA-DR or CD38 and Ki67 in T cells is related to a poor prognosis and greater disease severity in COVID-19 patients [24,47,48,49,50,51]. One of these studies suggested that the activation of T cells is higher in the lungs than in peripheral blood [47], while other studies identified two different cell subpopulations of activated CD8^+^ T cells in the PBMCs of COVID-19 patients: HLA-DR^+^CD38^dim^ cells, which are predominantly present in both mild and severe cases after 2 to 3 weeks of natural infection, and HLA-DR^+^CD38^high^ cells, which are only found in severe cases [49]. In line with these studies, we have seen that convalescent patients that presented high levels of CD8^+^ T-cell activation 4 weeks PI had clinical severity grades ranging from 4 to 6, meaning moderate to severe disease. In our studies, we observed that the activation of T cells was similar during the first 3 weeks for both naturally infected patients and vaccinees. This activation was maintained at week 4 after natural infection and then decreased during late convalescence. In contrast, vaccination did not increase T-cell activation beyond 4 weeks after receiving the first shot or even after a second or third vaccine shot. However, we have performed this analysis in the bulk CD8^+^ T-cell pool and not in SARS-CoV-2-specific T cells, which does not rule out the activation of bystander T cells.

These differences could be explained because of the exposure to different SARS-CoV-2 viral proteins when suffering from a natural infection compared to exclusive exposure to the spike (S) protein of the virus in individuals that have received the mRNA vaccine.

It has been demonstrated that activated CD8^+^HLA-DR^+^CD38^+^ T cells in a mild case of COVID-19 significantly expand following symptom onset, which reached their peak frequency of 12% of CD8^+^ T cells on day 9 after symptom onset and contracted thereafter [52]. Given the average time of 5 days from infection to the onset of symptom [53], the dynamics and magnitude of the T-cell response to SARS-CoV-2 is similar to those observed after immunization with live vaccines [54]. However, not much is known about how mRNA-based vaccines against SARS-CoV-2 [55,56] affect the dynamics of CD8^+^ T-cell activation marker expression and the memory profile. A study by Zhang et al. has recently shown that after two shots with the BNT162b2 vaccine (Pfizer/BioNTech), the percentage of CD8^+^ T cells expressing CD38 and HLA-DR increases, as well as the number of SARS-CoV-2-specific cells [57]. Another report found that following a primary vaccination and a booster, the expression of the CD38 and Ki67 markers increases but eventually declines [58]. Other studies have shown an induction of CD8^+^ T effector memory (EM) cells after vaccination, which contracted rapidly while central memory (CM) expanded. Moreover, spike-specific cells showed predominantly an EM phenotype [58,59].

We have also investigated the variation in CD8^+^ T-cell memory patterns between mRNA vaccinees and COVID-19 patients throughout this study. Upon comparing the proportion of EMRA, naive, EM, and CM CD8^+^ T cells across individuals experiencing natural infection and those who had received vaccination, we observed that during the late phase of the immune response, there were no differences on the levels of EM cells in the different groups (Figure 2D). However, CD8^+^ T EMRA cells were lower in the early convalescence phase which later increased, while vaccinees had a mild increase after the second vaccination dose, followed by a decrease after the third shot. Naive CD8^+^ T cells were present in low levels; however, they were higher in convalescent patients than in vaccinees, except for patients that received three vaccine shots, which showed the highest levels of naive T cells. Finally, we saw that patients in the early convalescent phase had the highest percentage of CD8^+^ CM T cells, which decreased during late convalescence, but they were still always higher than the levels of CM in vaccinees. In contrast to our results, other studies have shown that the levels of EMRA and EM cells were lower during the convalescence phase, and the relative number of CD8^+^ naive and CM cells was higher, compared to healthy individuals [58,60]. Another study found that convalescent patients had higher amounts of CD8^+^ T naive, EM, and CM cells when compared to individuals with delayed viral clearance [41]. Interestingly, Rajamanickam et al. saw that convalescent patients had an increase in the relative number of EMRA cells four months following the beginning of symptoms [61]. Other studies have shown the antiviral role of EMRA cells during HIV infection [62].

Publications in which vaccinated individuals were studied found that naive vaccinated individuals who were low responders to vaccination had higher EM cell levels, while high responders and convalescence had a higher frequency of CM cells [63]. Additionally, it was shown that immunization can alter the memory profile of S-reactive CD8^+^ T cells. Prior to immunization, a cohort of recovering patients presented EM, CM, and EMRA cells against SARS-CoV-2, which switched to only EM and EMRA cells after immunization [27,58]. In our analysis, we did not differentiate between high or low vaccine responders; however, we still saw a high percentage of EMRA cells after one (Vac-1x) or two (Vac-2x) vaccine doses. After the third shot (Vac-3x), there was a decrease in EMRA cells and an increase in naive T cells.

Divergent findings have been reported regarding TCR diversity in COVID-19 patients. While Schultheiß et al. [64] and Luo et al. [22] observed an increase during the disease and the onset of convalescence; Hou et al. [65] and Wang et al. [21] found that the diversity of TCRs decreased in COVID-19 patients (acute infection or convalescence). In our investigations, we observed that samples coming from both naturally infected patients and vaccinees had high TCR diversity, which was shown by a high Gini–Simpson score and high diversity in the length of the CDR3 region. In fact, a diverse length and sequence on the TCR CDR3 loops are related to the high clonality and ability of T cells to recognize a wide range of antigens [66,67]. Prior research has demonstrated that COVID-19 patients had a skewed distribution of the CDR3 length, and vaccine recipients and late convalescent patients may have larger CDR3 lengths as a result of different rearrangements [21,22,64,65]. Interestingly, when comparing the diversity among groups and even between patients in the same group, we saw no overlapping events, as shown by the Jaccard index. In addition, the variations on the use of the TRBV and J genes in the different experimental groups supported high TCR diversity. According to Hou et al. [65], there was a decrease in the use of the TRBV1, TRBV5, TRBV6, TRBV19, TRBJ1-4, and TRBJ2-6 genes in COVID-19 patients, while there was an increase in the use of genes from the TRBV3, TRBV9, TRBV11, TRBV12, TRBV15, TRBV21, TRBV24, TRBV27, and TRBJ1-6 families. In a study by Wang et al. [21], it was found that TRBV10 and TRBV2 had significantly lower expression in COVID-19 patients, while TRBV12, TRBV20, TRBV24, TRBV3, and TRBV9 showed higher expression in COVID-19 patients compared to healthy controls, whether in the baseline, acute, or convalescent group. In terms of TRBJ usage, TRBJ2-5 showed higher usage, while TRBJ1-3 and TRBJ1-4 showed significantly lower usage in COVID-19 patients when compared to healthy controls, whether in the baseline, acute, or convalescent stages. Conversely, Luo et al. [22] observed that convalescent patients used more TRBV20-1 and TRBJ2-7. Some of these data are in line with our results, such as the increase in TRBV24-1 in EC patients. These would suggest that the usage of the different genes might be determined by the individual genetic background as well as their exposure to previous coronavirus infections [68], supporting the theory that T cells linked to other coronaviruses drive the patient’s immune response in the early stages of infection.

In summary, our findings indicate the high diversity in the immune response to SARS-CoV-2 infection and vaccination, which was observed not only between experimental groups but also between individuals within the same group. Furthermore, our data suggest that SARS-CoV-2 natural infection would induce higher CD8^+^ T-cell activation during acute infection and early convalescence compared to vaccination. Moreover, T-cell activation did not seem to increase after the second and third vaccine shots. However, it has been widely shown that repetitive vaccination increases the levels of anti-SARS-CoV-2 antibodies, which is especially important in elderly and immune-suppressed individuals [11,69].

Finally, we provide a thorough summary of the TCRβ CDR3 repertoire in vaccination recipients and COVID-19 patients, which adds to the knowledge about T-cell diversity during SARS-CoV-2 infection and vaccination.

## Figures and Tables

**Figure 1 vaccines-13-00551-f001:**
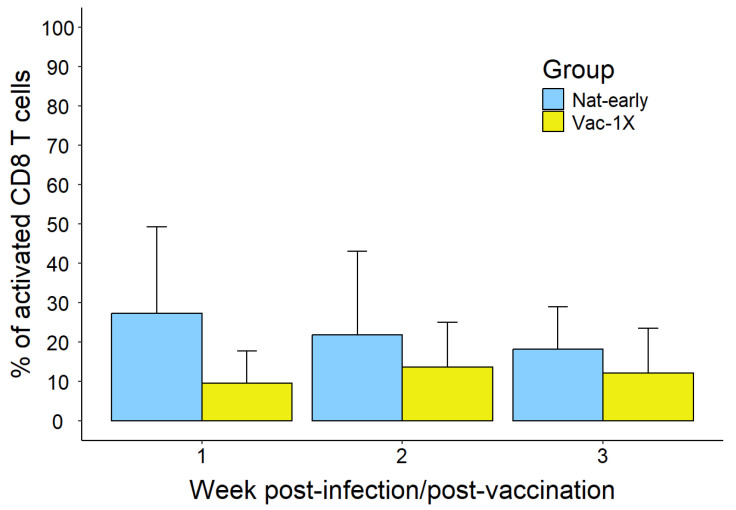
CD8^+^ T-cell activation during the early immune response in naturally infected and mRNA-vaccinated patients. Activation of bulk CD8^+^ T cells represented as percentage of CD8^+^CD38^+^HLA-DR^+^ in natural infection versus one vaccine dose. Total number of samples in each group: week 1: nNat-early = 5 and nVac = 2; week 2: nNat-early = 15 and nVac = 3; week 3: nNat-early = 5 and nVac = 6.

**Figure 2 vaccines-13-00551-f002:**
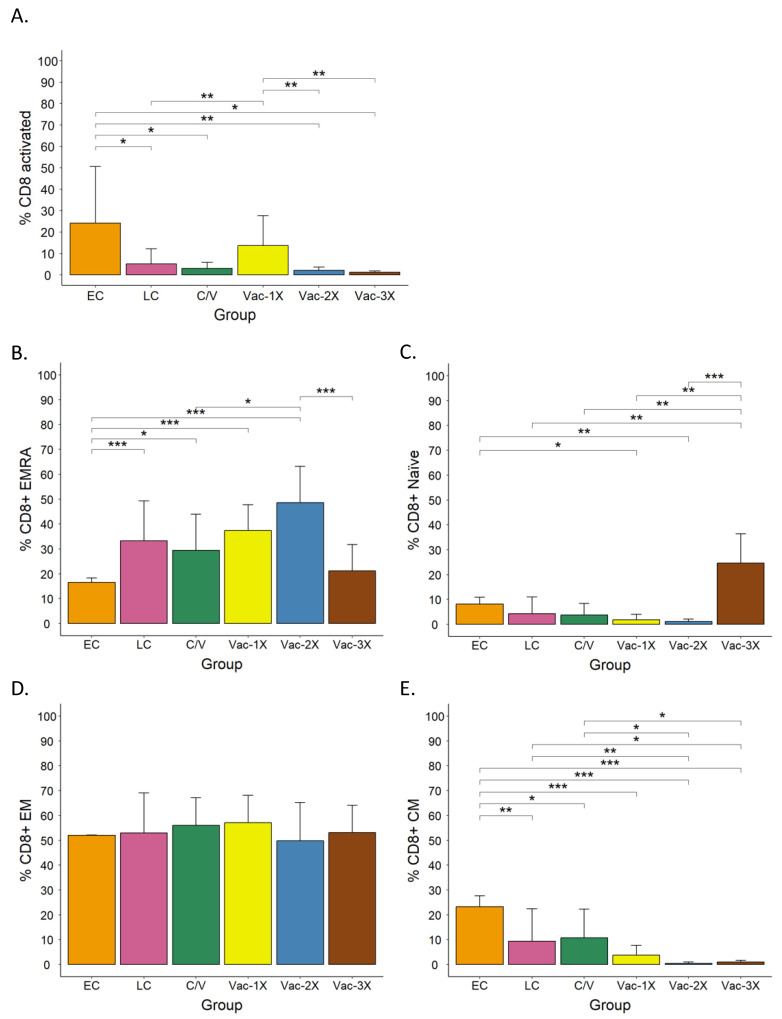
CD8^+^ T-cell activation and memory profiles in convalescent and mRNA-vaccinated patients 4 weeks after the onset of symptoms or vaccination. (**A**) Activation of bulk CD8+ T cells in early convalescence (EC), late convalescence (LC), convalescence-vaccinated (C/V), vaccinated with one (Vac-1X), two (Vac-2X) or three (Vac-3X) vaccine doses. (**B**) Percentage of CD8+ EMRA, (**C**) Naïve, (**D**) EM, and (**E**) CM T cells in the different groups. EMRA, effector memory cells re-expressing CD45RA; EM, effector memory; CM, central memory. A generalized least square (GLS) model with dummy variables, followed by Tukey’s multiple comparison test, was used for statistical analysis. * *p* < 0.05, ** *p* < 0.01, and *** *p* < 0.001.

**Figure 3 vaccines-13-00551-f003:**
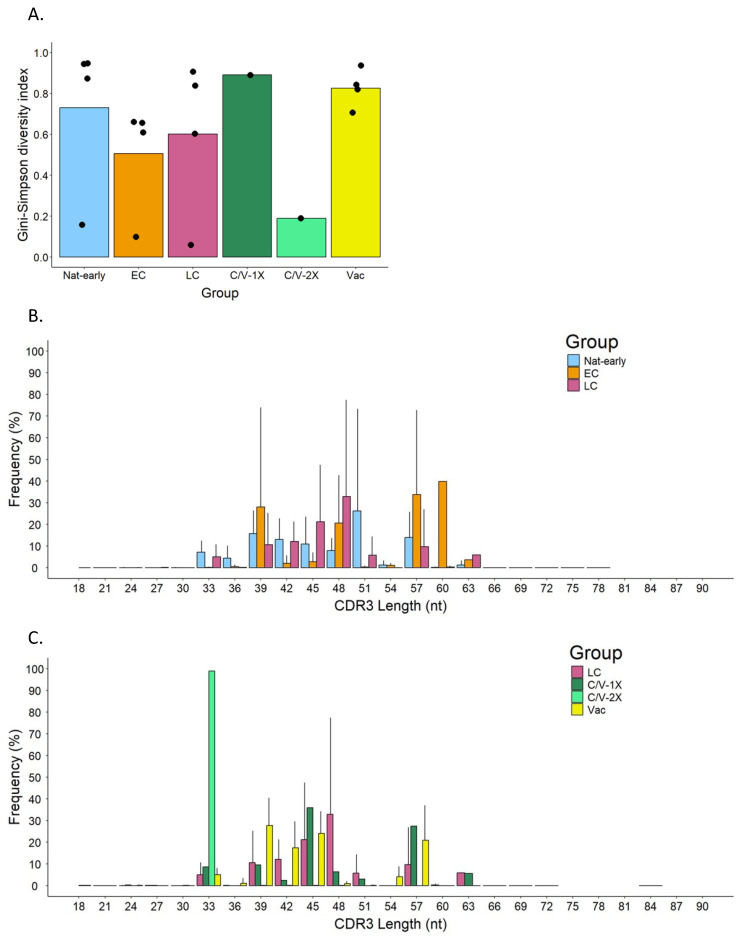
Diversity of the TCRβ repertoire in SARS-CoV-2-specific CD8^+^ T cells from convalescence patients and vaccinees. (**A**) The Gini–Simpson diversity index in acute infected (Nat-early), early (EC) and late (LC) convalescence, convalescence and vaccinated with one dose (C/V-1X), or two doses (C/V-2X) and vaccinated (Vac) individuals. The bar shows the GS mean between samples in each group. Black dots represent each patient. (**B**,**C**) Comparison of the CDR3 length frequency distribution between the different groups.

**Figure 4 vaccines-13-00551-f004:**
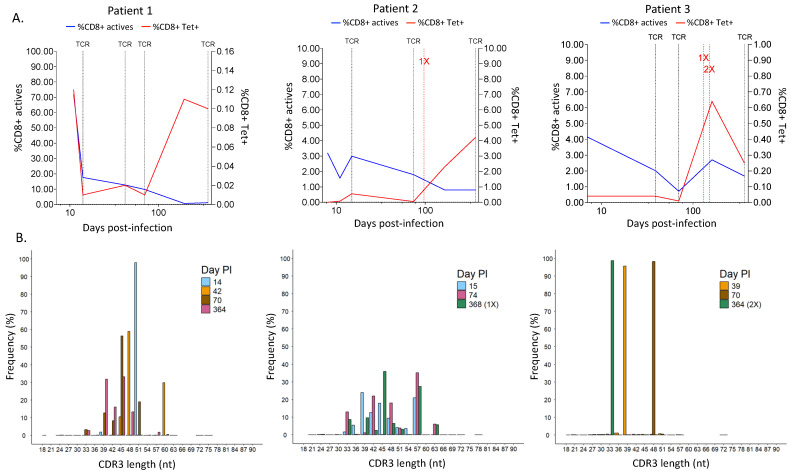
Comparison over time of the profiling of CD8^+^ T cells in the different patient samples. (**A**) Evolution along the time of the percentage of active CD8^+^ T cells and SARS-CoV-2-specific CD8^+^ T cells. Dotted black vertical lines in the graph mark the time points used for TCR sequencing. Patient 1: days 14, 42, 70, and 364 PI; Patient 2: days 15, 74, and 368 PI; Patient 3: days 39, 70, and 364 PI. Dotted red vertical lines mark the time point of the first vaccination (1X) and second vaccination doses (2X). (**B**) Frequency of different CDR3 length distributions in the number of nucleotides (nt) at a time point. Day PI = days post-infection. (**B**) CDR3 length distribution for different time points. Day PI = days post-infection. 1X = one vaccination dose, 2X = two vaccination doses.

**Figure 5 vaccines-13-00551-f005:**
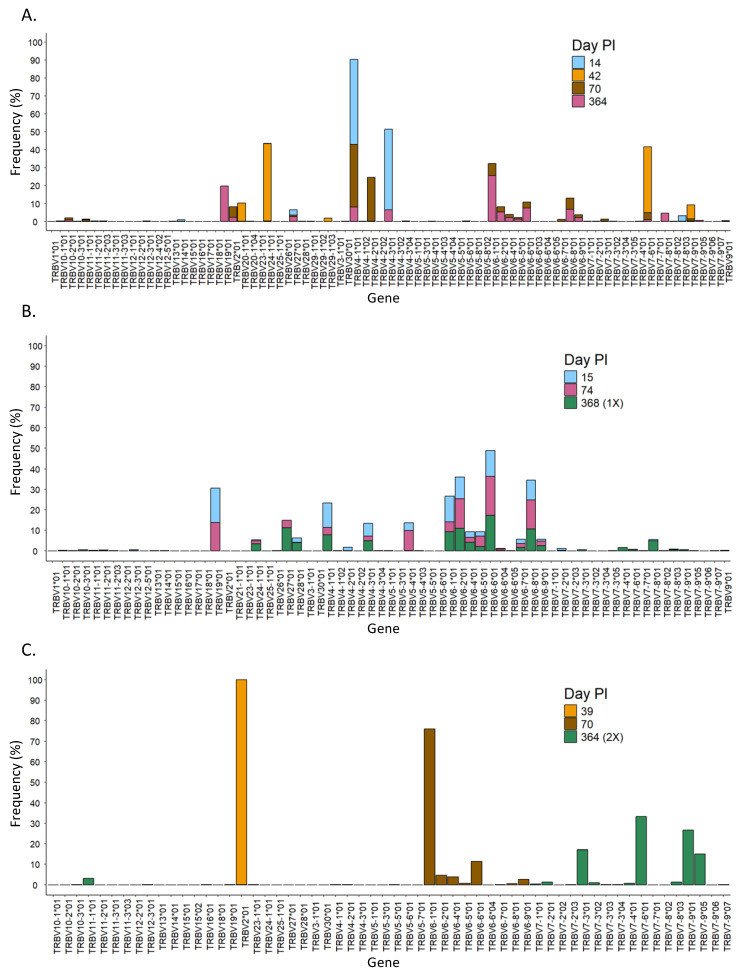
Evolution over time of the usage of the V gene in the different patients. Usage of TRBV genes at different time points in patients 1 (**A**), 2 (**B**), and 3 (**C**). PI = post-infection.

**Figure 6 vaccines-13-00551-f006:**
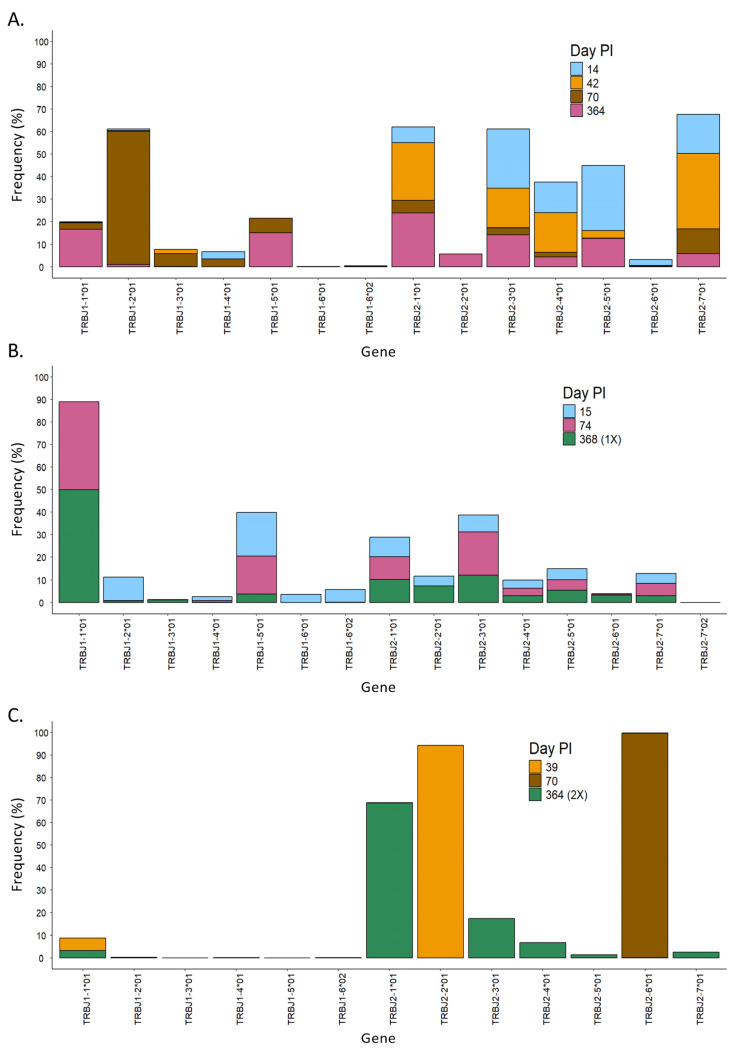
Evolution over time of the usage of the J gene in the different patients. Usage of TRBJ genes at different time points in patients 1 (**A**), 2 (**B**), and 3 (**C**). PI = post-infection.

## Data Availability

The data presented in this study are available on request from the corresponding author due to ethical reasons.

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
