# Peer review of "Longitudinal Immunoprofiling of the CD8+ T-Cell Response in SARS-CoV-2 mRNA Vaccinees and COVID-19 Patients"

_vaccines, 2025, doi:10.3390/vaccines13060551_

Round 1
Reviewer 1 Report
Comments and Suggestions for Authors
First, congratulations on completing the research. The search for an understanding of a part (CD8+ lymphocytes) of the cellular response to natural infection or through stimulation with the vaccine is important in understanding the mechanisms of the immune response to COVID-19.
I have only one suggestion in topic 2.1 (description of the cohorts).
I think the description of the cohorts could be better detailed. For example, what was the follow-up period for the two cohorts? How many samples were collected in total? Was the time from patient symptoms to seeking medical care taken into account? Could this make any difference in the assessment of the population profiles of CD8+ lymphocytes? Did all individuals admitted to the hospital have a laboratory diagnosis of COVID-19? How was this diagnosis made?
Author Response
Response: We thank the reviewer for the comments. We have now extended the information about the cohorts used in these studies. See section 2.1. Answers to specific questions are the following:
1. What was the follow-up period for the two cohorts? For the ECRIN cohort (Spain) patients were recruited from December 2020 to March 2021. For the BNITM cohort (Germany) samples were collected between May 2021 and January 2022.
2. How many samples were collected in total? For the ECRIN cohort (Spain), blood samples were obtained at the time of diagnosis and twice weekly during hospitalization. Upon discharge samples were obtained every 3 months for 18 months. From this cohort, we used samples from 21 different patients from which 2 to 5 samples were harvested overtime depending on the patient’s study commitment. A total of 87 samples were collected, and from those, 81 were used for the studies. For the BNITM cohort (Spain), samples were harvested one, two, three and four weeks upon de first vaccine shot, and then four weeks after the second and third shot. In total, 41 samples from 10 different volunteers were harvested and used in these studies.
3. Was the time from patient symptoms to seeking medical care taken into account? Yes. Patients reported suffering symptoms 2 to 10 days before hospital visit, depending on the case. Based on this and the incubation period for SARS-CoV-2, mean time for virus exposure was set at 7 days before hospital admission, which was defined as day 0 post-infection.
4. Could this make any difference in the assessment of the population profiles of CD8+ lymphocytes? Looking at individual patient data and their corresponding CD8+ profiles there was no correlation between the time between symptom onset and hospitalization, and the CD8+ profile.
5. Did all individuals admitted to the hospital have a laboratory diagnosis of COVID-19? Yes, the criteria to enter the cohort was to have a COVID-19 positive test and to have severe symptoms, such as respiratory distress and low oxygen saturation.
6. How was this diagnosis made? Diagnose was made by RT-PCR detection of the E gene in nasopharyngeal swabs and using the Panther Fusion Hologic (San Diego, CA, USA) automated molecular diagnostic platform, as previously described in Folgueira MD. et al. Prolonged SARS-CoV-2 cell culture replication in respiratory samples from patients with severe COVID-19. Clin Microbiol Infect. 2021 Jun;27(6):886-891. doi: 10.1016/j.cmi.2021.02.014. This information has been now added to the manuscript (see section 2.1).
Reviewer 2 Report
Comments and Suggestions for Authors
In this manuscript, Brunetti et al report their efforts in tracking and systemically analyzing the dynamic variations in CTLs response after being infected by SARS-CoV-2 and vaccinated using SARS-CoV-2 mRNA vaccines. The study is helpful for us to understand the dynamic T-cell immune response after SARS-CoV-2 infection and vaccination, also being beneficial for developing possible combinational strategies to boosting the anti-SARS-CoV2-2 infection, which is worth publishing in the journal. However, the following issues should be addressed before the manuscript to be accepted.
- The sample quantity in the study was relatively too small, such as in Figures 1 and 3. Are the results reliable enough?
- The significant differences between the indicated subgroups in Figure 2, especially for Figures 2B, 2C, and 2E seem to be exaggerated.
- The font size in the Figures 4, 5, and 6 is too small and difficult to be read clearly enough. The readability of these Figures should be improved.
- In Table 1, the authors only show which genes are upregulated or downregulated in different groups of COVID-19 and/or vaccinated patients. Is it possible to show the upregulated and/or downregulated levels using a heatmap?
Author Response
Response:
We thank you the reviewer for the helpful comments. We reply to the questions raised as follows:
- The sample quantity in the study was relatively too small, such as in Figures 1 and 3. Are the results reliable enough?
We agree with the reviewer that the number of samples available for this study was low. For this reason, in order to compare the activation and memory profiles between groups, we applied the GLS (Generalized Least Square) model with dummy variables followed by Tukey’s multiple comparison test as statistical analysis tests. This combination is a robust method to compare groups in which assumption of equal variances may not hold, which is commonly used in biomedical and clinical trial studies. The GLS model estimates parameters more efficiently when errors are not identically distributed, which is the case of our study. Moreover, the Tukey’s test is used to identified group pairs with statistically significant differences by comparing the means and not the standard deviations. After applying these methods to the data represented in Figures 1, the test gave no significant differences among groups.
In Figure 3A, we represent different analyses performed with the TCR sequencing data. The Gini Score shows a score between 0 and 1 that is assigned to each sample to grade its TCR diversity. In the graph, bars represent the mean score for the group and the black dots the score of each sample. In Figures 3B and 3C, we show the frequency and length the CDR3 sequences detected in the TCR sequencing. In summary, these are all approaches commonly used to analyze TCR sequencing data in a reliable manner.
- The significant differences between the indicated subgroups in Figure 2, especially for Figures 2B, 2C, and 2E seem to be exaggerated.
We thank the reviewer for pointing this out. We have now, indeed, reanalyzed the data including more samples. After this new analysis, we obtained lower activation percentage of the CD8+ T cells in the EC group. However, the differences between groups are still statistically significant when we apply a GLS model followed by a multiple comparison Tukey’s test, as explained in the response above.
Text through the manuscript (see highlighted text in yellow) has now been adapted to the new activation values.
- The font size in the Figures 4, 5, and 6 is too small and difficult to be read clearly enough. The readability of these Figures should be improved.
This has now been improved as suggested.
- In Table 1, the authors only show which genes are upregulated or downregulated in different groups of COVID-19 and/or vaccinated patients. Is it possible to show the upregulated and/or downregulated levels using a heatmap?
Thank you for the suggestion. We have realized that the data in Table 1 belonged to an old version of the manuscript and it is redundant with the data represented in supplementary figures 2 and 3. We apologize about this. We have now removed the table and any mention to it in the text and left the supplementary figures.
Reviewer 3 Report
Comments and Suggestions for Authors
The authors presented an interesting study on the CD8+ T cell response to SARS-CoV-2 infection and vaccination. In this study, they characterised the clonality and diversity of T cell receptors (TCRs) in patients at various stages after infection and vaccination.
Some comments are provided below.
1. The description of the study groups is unclear. Please pay closer attention to the description of the study cohorts. Unfortunately, the study does not include a group of naive, unvaccinated patients. Which SARS-CoV-2 strains were circulating when the biological samples were collected?
2. Please use 'naive' instead of 'naïve'.
3. Lines 67–71. This paragraph is not related to the study's topic and can be removed from the manuscript.
4. Lines 72–93: The description of T cell and antigen interactions should be expanded. How does the TCR interact with the antigen? How is the antigen-MHC complex formed?
5. Lanes 88–90. Describe how the TCR repertoire is modulated during SARS-CoV-2 infection.
6. Line 162. Please provide primer sequences.
7. Lines 213–222: The study cohorts are unclear. Which patients were classified as having an early immune response and which as having a late immune response? For vaccinated patients, was the time since vaccination taken into account? Vaccinated patients may also exhibit both early and late immune responses.
8. Line 229. Please explain how the percentage of activated CD8+ cells was calculated. What was set as 100%?
9. Lines 245–251: Additional study groups are described here. Please provide a description of all study cohorts at the beginning of the 'Results' section.
10. Line 274. Please clarify what the presence of the CD45RA and CCR7 markers indicates. Why are these markers important in the immune response to the virus?
11. Lines 311–312: Why was epitope YLQPRTFLL chosen?
12. Figure 4A. Please indicate the time points at which the analysis was performed.
Author Response
Response:
- The description of the study groups is unclear. Please pay closer attention to the description of the study cohorts. Unfortunately, the study does not include a group of naive, unvaccinated patients. Which SARS-CoV-2 strains were circulating when the biological samples were collected?
We have now expanded and improved the description of the cohorts in section 2.1.
We agree with the reviewer that it is unfortunate not to have included samples from naïve and unvaccinated individuals. At the time when sample collection started (March 2021) was already very difficult to find this kind of samples and, therefore, we could not include them in our studies.
Most of the patients were infected with the ancestral SARS-CoV-2 (Wu-hu-1) or the VoC Alpha (B.1.1.7).
- Please use 'naive' instead of 'naïve'.
This has been changed across the manuscript (see highlighted text through).
- Lines 67–71. This paragraph is not related to the study's topic and can be removed from the manuscript.
As suggested, this paragraph has been removed.
- Lines 72–93: The description of T cell and antigen interactions should be expanded. How does the TCR interact with the antigen? How is the antigen-MHC complex formed?
As suggested, this part has now been expanded, see lines 63 to 73.
- Lanes 88–90. Describe how the TCR repertoire is modulated during SARS-CoV-2 infection.
As suggested, this part has now been extended, see lines 86 to 92.
- Line 162. Please provide primer sequences.
Primer sequences have been already published in reference 17. We have now included a sentence referencing to it (see line 185-186).
- Lines 213–222: The study cohorts are unclear.
The description of the different cohorts has been now improved and extended in section 2.1
Which patients were classified as having an early immune response and which as having a late immune response?
Samples obtained in the first 3 weeks after natural infection or after the first vaccine shot were classified as samples belonging to the early immune response. Samples harvested 4 weeks or later after natural infection or the first, the second or the third vaccine shot were classified as belonging to the late immune response. This classification was based on what previously described in references 21-23 and 38-40.
For vaccinated patients, was the time since vaccination taken into account? Vaccinated patients may also exhibit both early and late immune responses.
Yes, it was taken into account for samples obtained from vaccinees, similar to naturally infected individuals.
- Line 229. Please explain how the percentage of activated CD8+ cells was calculated. What was set as 100%?
The percentage of activated CD8+ cells was calculated as the percentage of cells expressing both CD38 and HLA-DR in the total CD8+ T cell population (100%).
- Lines 245–251: Additional study groups are described here. Please provide a description of all study cohorts at the beginning of the 'Results' section.
We have now expanded the information about the cohorts in section 2.1. Besides, we have added there a paragraph explaining how samples were divided in two groups based on the early or late immune response. Furthermore, deeper description of how each of these two groups (early and late immune response samples) were subdivided in other groups (Nat early, Vac1x, Vac2x, EC, LC, C/V, etc…) are kept in each section of the results (sections 3.1 and 3.2, respectively). We thank the reviewer for pointing this out; however, we respectfully disagree with the reviewer in regard to providing a description of all cohorts at the beginning of the Results section, we think that for means of understanding it is better to keep the description of the subgroups in their corresponding sections.
- Line 274. Please clarify what the presence of the CD45RA and CCR7 markers indicates. Why are these markers important in the immune response to the virus?
This has now been clarified and extended, see lines 289-305.
- Lines 311–312: Why was epitope YLQPRTFLL chosen?
The epitope YLQPRTFLL is derived from the viral spike, is recognized by public CDR3 motifs and it has been shown to be the most immunogenic peptide from SARS-CoV-2, see reference 43.
- Figure 4A. Please indicate the time points at which the analysis was performed.
Time points information has been now added to the figure legend.
Round 2
Reviewer 2 Report
Comments and Suggestions for Authors
All my concerns have been addressed appropriately, and I recommend accepting the manuscript for publication in the journal as is.